# Increased Expression of *VANGL1* Is Predictive of Lymph Node Metastasis in Colorectal Cancer: Results from a 20-Gene Expression Signature

**DOI:** 10.3390/jpm11020126

**Published:** 2021-02-14

**Authors:** Noshad Peyravian, Stefania Nobili, Zahra Pezeshkian, Meysam Olfatifar, Afshin Moradi, Kaveh Baghaei, Fakhrosadat Anaraki, Kimia Nazari, Hamid Asadzadeh Aghdaei, Mohammad Reza Zali, Enrico Mini, Ehsan Nazemalhosseini Mojarad

**Affiliations:** 1Basic and Molecular Epidemiology of Gastrointestinal Disorders Research Center, Research Institute for Gastroenterology and Liver Diseases, Shahid Beheshti University of Medical Sciences, Tehran 19875-17411, Iran; n.peyravian@gmail.com (N.P.); zahrapezeshkian@yahoo.com (Z.P.); ol.meysam92@gmail.com (M.O.); kavehbaghai@gmail.com (K.B.); k.nazari.1389@gmail.com (K.N.); hamid.assadzadeh@gmail.com (H.A.A.); 2Department of Neurosciences, Imaging and Clinical Sciences, “G. D’Annunzio” University of Chieti-Pescara, 66100 Chieti, Italy; stefania.nobili@unich.it; 3Department of Pathology, Shohada Hospital, Shahid Beheshti University of Medical Sciences, Tehran 19875-17411, Iran; afshinmo2002@gmail.com; 4Colorectal Division of Department of Surgery, Taleghani Hospital, Shahid Beheshti University of Medical Sciences, Tehran 19875-17411, Iran; dr.anaraki47@gmail.com; 5Gastroenterology and Liver Diseases Research Center, Research Institute for Gastroenterology and Liver Diseases, Shahid Beheshti University of Medical Sciences, Yaman Street, Chamran Expressway, Tehran 19857-17411, Iran; nn.zali@hotmail.com; 6Department of Health Sciences, University of Florence, Viale Pieraccini 6, 50139 Firenze, Italy

**Keywords:** colorectal cancer, gene signature, mRNA expression, *VANGL1*, FFPE

## Abstract

This study aimed at building a prognostic signature based on a candidate gene panel whose expression may be associated with lymph node metastasis (LNM), thus potentially able to predict colorectal cancer (CRC) progression and patient survival. The mRNA expression levels of 20 candidate genes were evaluated by RT-qPCR in cancer and normal mucosa formalin-fixed paraffin-embedded (FFPE) tissues of CRC patients. Receiver operating characteristic curves were used to evaluate the prognosis performance of our model by calculating the area under the curve (AUC) values corresponding to stage and metastasis. A total of 100 FFPE primary tumor tissues from stage I–IV CRC patients were collected and analyzed. Among the 20 candidate genes we studied, only the expression levels of *VANGL1* significantly varied between patients with and without LNMs (*p* = 0.02). Additionally, the AUC value of the 20-gene panel was found to have the highest predictive performance (i.e., AUC = 79.84%) for LNMs compared with that of two subpanels including 5 and 10 genes. According to our results, *VANGL1* gene expression levels are able to estimate LNMs in different stages of CRC. After a proper validation in a wider case series, the evaluation of *VANGL1* gene expression and that of the 20-gene panel signature could help in the future in the prediction of CRC progression.

## 1. Introduction

Radical surgery and adjuvant chemotherapy improve the clinical outcome of stage III and high-risk stage II colorectal cancer (CRC) patients. However, it is known that 5-year overall survival (OS) highly varies according to important prognostic factors, such the pTs stage (stages II and III) and the involvement of lymph nodes (pN1 and pN2 stage III) [1,2,3]. Overall, lymph node metastasis (LNM) is a key prognostic factor for the determination of CRC outcomes and significantly relates to poorer prognosis, disease-free survival (DFS), and OS [4,5]. 

Available clinical data suggest that the accurate diagnosis of LNM is not only important for the prediction of the prognosis of patients but also useful for further therapeutic management, such as for the selection of patients who would benefit from adjuvant/neoadjuvant chemotherapy or chemo-radiotherapy [2,6,7]. In fact, the number and sites of lymph nodes involved have a direct impact on the stage of disease as established by the AJCC tumor-node-metastasis (TNM) staging classification for colon cancer [8]. 

However, it should be noted that pathological methods are not able to diagnose occult LNMs (micrometastases). Thus, the use of advanced methodologies (e.g., gene expression profiling (GEP)), the investigation of specific biomarkers (e.g., microsatellite instability (MSI)), CpG island methylator phenotype (CIMP), the application of the immune score recommended for increasing the detection power of disease recurrence, and in this regard the use of genes associated with lymph node involvement are very useful to enhance this evaluation [9,10,11,12]. In this regard, GEP has been used to discover biomarkers associated with lymph nodes in epithelial neoplasms, such as pancreatic cancer [13], oral squamous cell carcinoma [14], invasive breast cancer [15], and CRC [16]. However, the power of diagnosis may vary based on gene selection.

High-throughput studies in which biomarkers of tumor suppressor genes and oncogenes are potentially able to predict the prognosis of CRC patients at different stages of disease and according to the lymph node involvement are available [4,8,17].

In order to find a suitable biomarker to predict LNM involvement, we evaluated gene expression profiling studies and selected 20 genes (*VANGL1*, *SMAD2*, *BUB1*, *EGFR*, *HES1*, *MAP2K1*, *NOTCH1*, *ANXA3*, *SMAD4*, *MTA1*, *LEF1*, *RHOA*, *TGF-ß*, *CD44*, *CD133*, *IL2RA*, *IL2RB*, *PITX2*, *PCSK7*, and *FOLH1*) that play a key role in carcinogenesis, tumor growth, LNM development tumor invasion, and metastasis by regulating a variety of cellular processes (Table 1) [4,16,17,18,19,20]. 

Thus, the aim of this study was to identify at the mRNA level and to validate at the protein level the potential prognostic role of these candidate genes in relation to the LNM of CRC patients.

## 2. Materials and Methods

### 2.1. Patients and Sample Collection

This retrospective study was performed in 100 formalin-fixed paraffin-embedded (FFPE) tumor tissues of CRC patients at stage I or II (*n* = 52) and at stage III or IV (*n* = 48) and 10 FFPE samples of normal tissues (colonic mucosa) as calibrators in RT-qPCR, as well as paired samples of normal colonic tissues from the same CRC patients. All samples were anonymized. Patients underwent surgical resection for CRC from February 1998 to December 2018 at Taleghani Hospital, Shahid Beheshti University of Medical Sciences, Tehran, Iran. They were chemo- and radiotherapy naïve, and none of them experienced previous neoplastic disease. Clinical information, such as colonoscopy/pathology report, follow-up data, and cause of death, was collected from medical records. All patients were carefully followed up to confirm their clinical outcomes.

This study was approved by the ethical committee (IR.SBMU.RIGLD.REC.1396.947) of the Research Institute for Gastroenterology and Liver Disease, Shahid Beheshti University of Medical Sciences, Tehran, Iran. Written informed consent was obtained from all patients. The inclusion criteria for the patients were the following: (1) signed informed consent; (2) availability of the pathology report to confirm the tumor histology; (3) nonresident in an institution, such as a prison, nursing home, or shelter; (4) no severe illness in the intensive care unit; and (5) no preoperative chemotherapy and radiotherapy. The exclusion criteria were the following: patients affected by familial adenomatous polyposis (FAP), hereditary nonpolyposis CRC (HNPCC), cancer at any site at the time of selection, and patients who received neoadjuvant chemotherapy or radiotherapy. The FFPE tissue blocks were cut 10–15 μm and 4–7 μm in thickness for mRNA extraction and immunohistochemistry (IHC), respectively.

To ensure the quality of the presence of tumor and normal cells in FFPE tissue blocks, before performing the laboratory process, each section was evaluated for tumor and normal cells (>80% representative) by the pathologist using hematoxylin and eosin (H&E) staining. 

### 2.2. RNA Isolation

Ten–fifteen micrometer thick sections were cut from the FFPE blocks, and each section was transferred into a microcentrifuge tube. Deparaffinization was performed with 1 mL xylene, incubating twice for 10 min, and 1 mL absolute ethanol, also incubating twice for 10 min. 

### 2.3. Quantitative and Qualitative Analysis of the Isolated RNA Samples

Total RNA was extracted from the target tissues using the Rneasy Kit (Qiagen, Chatsworth, CA) according to the company’s protocol. To avoid genomic DNA contamination, RNA samples were treated with Dnase I according to the manufacturer’s protocol (Invitrogen, Carlsbad, CA, USA). RNA concentration was measured by a NanoDrop ND-1000 spectrophotometer (NanoDrop Technologies Inc., Rockland, DE, USA). An A260/A280 ratio was used to evaluate the RNA purity, and values in the range of 1.8–2.0 were accepted. 

### 2.4. Real-Time PCR Analysis

cDNAs were generated with a PrimeScript RT Reagent kit (Takara, Shiga, Japan) according to the manufacturer’s protocol.

The mRNA levels of 20 candidate genes (i.e., *VANGL1*, *IL2RA*, *IL2RB*, *TGF-ß*, *SMAD2*, *SMAD4*, *CD44*, *CD133*, *HES1*, *NOTCH1*, *LEF1*, *MTA1*, *EGFR*, *MAP2K1*, *FOLH1*, *BUB1*, *RHOA1*, *PCSK7*, *PITX2*, and *ANXA3*) (Table 1) and of the housekeeping gene β2-microglobulin (*B2M*) were analyzed by RT-qPCR using the SYBR Fast qPCR Mix Kit (Takara). The cDNA samples were amplified by the 7500 Real-Time PCR System (Applied Biosystems, Foster City, CA, USA) with an initial denaturation at 95 °C for 30 s, followed by 40 cycles each at 95 °C for 5 s and 60 °C for 34 s. Relative expression abundances of the target genes were determined by normalizing to *B2M* and *β-actin* using the 2^−ΔΔCt^ method. Each measurement was performed in triplicate. *B2M* was utilized to calculate the relative quantitation (RQ) of mRNA transcripts using the 2^−ΔΔCt^ method. 

### 2.5. Unsupervised Hierarchical Clustering 

An unsupervised hierarchical clustering was used to graphically display the expression levels of 20 candidate genes in CRC samples. Dendrograms and clustering were generated by using the Gene Cluster version 3.0 software and visualized with the Java TreeView version 3.0, available at http://rana.lbl.gov/EisenSoftware.htm (accessed on 1 February 2021) and http://jtreeview.sourceforge.net (accessed on 1 February 2021), respectively. The color of each square box represents the ratio of gene expression. Green boxes indicate upregulated genes, while red boxes represent downregulated genes.

### 2.6. Immunohistochemistry and Evaluation of Staining

To investigate the expression levels of candidate proteins, an IHC analysis was performed on slices of FFPE tissues ranging from 4 to 7 µm in thickness. For deparaffinization, the slides were incubated at 37 °C for 24 h and then washed with xylene (100%), ethanol (100%, 85%, and 75%), and distilled water, respectively. After deparaffinization, slides were incubated in a solution of 10% H_2_O_2_ and methanol at a ratio of 1:9 for 15 min and subsequently washed with the distilled water. Next, the slides were treated in the 10 mM citrate buffer solution (pH = 6) and microwaved with 800 W for 24 min and washed with the Tris-buffered saline (TBS). After treating with the blocking serum for 15 min, the slides were immunostained with mouse anti-human MoAbs for 45 min and later washed with TBS. Later, by treating with the EnVision + visualization system (Dako) for 30 min, followed by DAB (Master Diagnosis, LOT. No 090517C1-01) as the chromogen substrate for 10 min, the bound primary antibody was visualized. Finally, the slides were washed with distilled water, dehydrated with ethanol, and stained in hematoxylin. All the slides were independently checked by investigators who had no knowledge of the patients’ characteristics and clinical outcome using a microscope (Nikon, Tokyo, Japan). 

The analysis of immunostaining intensity was performed using a qualitative scale and ocular observation. Sections were first scanned at low-power magnification (10x) and were quantitatively assessed as follows: under a light microscope at 400x magnifications, five high-power fields (HPFs) were randomly selected, and the immunostaining intensity was determined. 

Mean values were estimated through the scanning of the entire tissue sections of all samples using two graded scales: negative, <10%, and positive, >10%. The positive controls were the following: (a) a normal colonic tissue was taken as an internal control for ß-catenin IHC, and (b) a histologically diagnosed section of colon carcinoma tissues for nuclear positivity by ß-catenin IHC. Negative control was achieved by omitting the primary antibody. The MoAbs used in this study were VANGL1 (Abcam, Anti-VANGL1 antibody ab69227), SMAD4 (Abcam, Phospho-SMAD4 antibody T277), EGFR (Master Diagnosis, Anti-EGFR antibody Lot No. 0664000), and LEF1 (Master Diagnosis, Anti-EGFR antibody Lot No. 07430003).

### 2.7. Statistical Analysis

Tumors were divided into lymph node metastases (LNMs) and non-lymph node metastases (non-LNMs) based on the histopathological results. The mRNA expression levels of tumor tissues were represented as the mean ± standard deviation (SD). The Mann–Whitney U and Kruskal–Wallis tests were used to assess the differences of the mRNA expressions of 20 genes between the established groups (i.e., presence/absence of LNMs; stage (I–II vs. III–IV), tumor differentiation grade (well vs. poor differentiated), sex (male vs. female), and age (<50 vs. ≥50)). All statistical analyses were performed by the IBM SPSS Statistics software version 22 (IBM, SPSS, Chicago, IL, USA) and Stata analyzer. 

For receiver operating characteristic (ROC) curve analysis, the R 3.6.1 software was used to evaluate the sensitivity and specificity of the prognosis prediction (evaluated by OS) according to the mRNA gene expression by analyzing the area under the curve (AUC). Stratification of patients in high and low tumor gene expression was established according to the cutoff obtained for each gene by ROC analysis. OS analysis was performed by plotting Kaplan–Meier (log-rank test) curves. *p*-Values < 0.05 were considered statistically significant.

## 3. Results

### 3.1. Clinical and Pathological Characteristics of Patients

The population study consisted of 100 FFPE tissues from CRC patients (59 men and 41 women with an average age of 52.17 years, 20–78 range). The clinical features of the study population are shown in Table 2. Information on age, sex, stage, tumor differentiation, and tumor location is available for all patients. Among patients, 52% had stage I or II CRC, while 48% of the cases had stage III or IV. Of 100 patients, 37 were positive and 63 were negative for LNM.

### 3.2. Gene Expression Analysis

To identify molecular determinants of LNMs, gene expression profiles from patients with or without LNMs and at different stages of disease were compared. Based on literature data, we selected 20 genes that relate to the lymphatic metastatic process and evaluated their expression levels in 100 FFPE blocks.

Relationships of tumor gene expression with demographic (sex, age), clinical (tumor location), and pathological (stage, LNM, grade) features are reported in Table 3 and Table 4. 

In particular, the gene expression levels of *VANGL1* varied significantly between patients with and without LNMs. The tumors of patients with LNMs displayed twofold higher levels of *VANGL1* mRNA expression compared with those of patients without LNMs (*p* = 0.02) (Table 4 and Figure 1). Additionally, the expression levels of this gene varied between patients with stages I-II and III-IV, showing the highest mean level (i.e., 8.831) for stages III and IV. This difference reached a good level of significance, although not fully significant (*p* = 0.05) (Table 4 and Figure 1).

The mRNA expression levels of three genes (i.e., *IL2RB*, *SMAD*, and *ANXA3*) were significantly (*p* < 0.05) different between well-differentiated (i.e., G1 and G2) and poorly differentiated (i.e., G3 and G4) tumors (Table 4 and Figure 2). 

We found significant associations between tumor mRNA expression of *IL2RB* and *NOTCH1* genes and gender and between tumor mRNA expression of *IL2RA* and *MAP2K1* genes and age. In particular, the *IL2RA* gene was significantly downregulated in patients younger than 50 years old compared with patients older than 50 years (Figure 3). Additionally, an increased expression of the *MAP2K1* gene was observed in patients older than 50 years in comparison with patients younger than 50 years old (*p* = 0.02). *IL2RB* exhibited lower expression in females compared with males (*p* = 0.02) (Table 3 and Figure 4). Conversely, the *NOTCH1* gene was significantly upregulated in female patients as compared with male CRC cases (*p* = 0.02).

### 3.3. Heat Maps of Real-Time PCR Data 

Hierarchical clustering of 100 CRC samples is reported in Figure 5. According to the diagram, the *VANGL1*, *PCSK7*, and *ANXA3* genes showed the highest expression levels in most CRC samples.

### 3.4. ROC Analysis

The predictive performance of the 20-gene signature was assessed by computing the AUC value of the ROC curve. A logistic regression model was built based on the comparison of tumor samples (*n* = 100) in relation to the following study patient characteristics: stage I–II vs. III–IV and presence vs. absence of LNM. We selected two panels including 5 and 10 genes based on the genes that had the highest AUC and showed a more effective role in CRC progression. One panel included 5 genes (*VANGL1*, *IL2RA*, *SMAD2*, *RHOA1*, and *HES*), and the other 10 genes (previous 5 plus *MTA1*, *CD133*, *FOLH1*, *NOTCH1*, and *TGF-ß*). The total number of genes (i.e., 20-gene panel) was also analyzed. 

Figure 6 summarizes the performances of the study gene panels for the prediction of stage in the patient cohort, with the 20-gene panel achieving the highest performance. The AUC value for the 5-gene panel was 68.39%, along with 95% CI, 57.81%–78.97%; 67.30% sensitivity; and 66.66% specificity (Figure 6A); for the 10-gene panel, the AUC was 71.67% (95% CI, 61.51%–81.84%; sensitivity, 61.53%; and specificity, 72.91%) (Figure 6B). The analysis of the total 20 genes resulted in AUC = 78.85% (95% CI, 69.94%–87.75%; sensitivity, 75%; and specificity, 77.08%) (Figure 6C). In Figure 6D, the AUCs of 20-, 10-, and 5-gene panels in relation to stage are reported. When all the three AUCs (5-/10-/total-gene panels) were compared together, these results showed a trend towards significance (*p* = 0.055). When the AUC of the total number of genes was compared with that of the 5 genes, the difference was highly significant (*p* = 0.02). A statistical trend was observed between the AUC of the total panel and that of the 10-gene panel (*p* = 0.08), while no significant difference between the AUC of the 10-gene panel and that of the 5-gene panel was noted (*p* = 0.34). 

Figure 7 summarizes the performances of the study gene panels for the prediction of LNMs in CRC patients, and also in this case, the 20-gene panel achieved the highest performance. The AUC value was 70.19% (95% CI, 59.18%–81.02%; sensitivity, 84.12%; specificity, 57.86%) when the gene expression of the 5-gene panel was compared in relation to LNM and non-LNM CRC patients (Figure 7A). Comparison of the gene expression of the 10-gene panel in relation to LNM and non-LNM CRC patients resulted in AUC = 71.47% (95% CI, 60.62%–82.32%; sensitivity, 80.95%; specificity, 59.45%) (Figure 7B). The AUC of the total genes in LNM vs. non-LNM CRC patients was the highest of the three AUCs obtained (i.e., 79.84% (95% CI, 70.38%–89.30%; sensitivity, 74.60%; specificity, 75.67%) (Figure 7C). As far as the association with LNMs was concerned, the comparison between all the AUCs together (5-/10-/total-gene panels) pointed out a nearly significant difference (*p* = 0.05). In particular, the AUC of the total-gene panel was significantly higher compared with that of the 5-gene panel (*p* = 0.03) in relation to LNM and non-LNM CRC patients (Figure 7D). A high statistical trend was observed between the AUC of the total gene panel and that of the 10-gene panel (*p* = 0.06), although no statistical difference was observed between the AUC of the 10-gene panel and that of the 5-gene panel (*p* = 0.38). We also analyzed the predictive performance of single genes according to stage and LNM (Appendix A). Data showed that the *VANGL1* gene was a significant predictor for LNMs with an AUC of 63.99 (95% CI, 52.41%–75.56%; sensitivity, 80.95%; specificity, 45.94).

### 3.5. Correlation of Gene Expression with Overall Survival

All patients completed their follow-up by 20 December 2018 (median, 10.8 years, and range, 0.019–21 years). Patients whose tumors expressed higher levels of *NOTCH1* mRNA or lower levels of *IL2RB* mRNA showed a statistically significant prolonged OS compared with their respective counterparts (*p* = 0.042 and *p* = 0.043, respectively) (Figure 8). No other statistically significant correlation was found between OS and expression levels of the other study genes.

### 3.6. Immunohistochemistry Analysis

The protein expression levels of four genes that play a critical role in cancer development and progression was evaluated by using IHC. Twenty-five percent of CRC FFPE and normal matched tissues were used in this regard. In particular, we were interested in the evaluation of the expression levels of the products of *VANGL1*, *EGFR*, and *SMAD4* based on literature data and on the relationships we observed between the expression of their respective encoding genes and the clinical/pathological characteristics of the study CRC patients. The fourth protein we selected was LEF1. Although we did not find relationships between *LEF1* gene expression and the clinical/pathological parameters of CRC patients, we were interested in evaluating the potential role of LEF1 as an early biomarker of colorectal carcinogenesis since its activation by MYC has been associated with the activation of the WNT pathway signaling. The protein expression of VANGL1, EGFR, LEF1, and SMAD4 via their antibodies were examined. Results showed that VANGL1 and EGFR proteins were overexpressed (more than 50% of the stained cells in colon adenocarcinoma tissues compared with the normal tissues) (Figure 9 and Figure 10). Additionally, immunohistochemical staining revealed a predominantly nuclear localization of SMAD4 and LEF1, and they showed higher expression in CRC tissues compared with normal colonic mucosa with more than 50% and 20% of the stained cells, respectively (Figure 10 and Figure 11).

## 4. Discussion

The prediction of CRC progression risk and the identification of novel biomarkers predictive of this risk could represent a relevant advancement [6]. In this study, we evaluated the expression levels of several genes that are involved in LNM and malignant transitions in CRC tissues via RT-qPCR and IHC methods. 

Previous investigations showed a high tumor expression level of the *VANGL-1* gene in CRC patients compared with normal tissues [21,22,23,24]. Additionally, *VANGL1* gene expression levels have been suggested to play a critical role in CRC progression and to be notably related to tumor stages and LNM [21,22,23,24]. These findings are in substantial agreement with our results. Lee and et al. showed that *VANGL1* gene knockdown can decrease the mRNA expression level of *CYKLIND1*, *COX2*, *MMP3*, and *ERK1/2* and reduce tumor growth and invasion. In addition, they found that a high expression level of the *VANGL1* gene was associated with the overexpression of *AP-1* target genes, which have an important role in MAPK signaling in CRC [22]. Oh et al. indicated that *VANGL1* silencing reduced vascular endothelial growth factor A (VEGF-A) and hypoxia-inducible factor 1-alpha (HIF1A). They suggested that the *VANGL1* gene can increase angiogenesis and CRC malignancy [21]. Additionally, our past investigations showed that angiogenesis and the angiogenic factors *VEGF-A* and *HIFA* play an important role in CRC initiation and progression [25,26,27]. Thus, it seems that the *VANGL1* gene may interact with VEGF-A and HIF1A signaling and enhance tumor malignancy.

In this study, we showed that the *VANGL1* gene was an independent prognostic biomarker for CRC patients. Taken together, these results indicate that the *VANGL1* gene may have a key role in the regulation of several genes, including those involved in angiogenesis. Thus, *VANGL1* could be suggested as a potential biomarker for the prediction of tumor malignancy and targeted therapy in CRC.

*TGF-β* has been suggested to be a tumor suppressor gene able to stop the cell cycle at early stages of tumor, and SMAD proteins, being transcriptional mediators of TGF-β signaling, play a critical role in it [28]. In particular, *SMAD2* is located at 18q21 and plays a role as a tumor suppressor gene [29,30,31]. As a result of the loss of heterozygosity (LOH) in the 18q21 region, *SMAD2* gene expression is reduced in several cancers and increases cancer progression [31]. However, our findings showed a significant downregulation of *SMAD2* in well-differentiated tumors. Although this finding is substantial in contrast with most of the available data, the complexity of the mutational profile of *SMAD2* [32], its relationships with the other SMAD proteins [33], and the potential role of specific miRNAs on the regulation of *SMAD2* [34] could have contributed to this result. 

*SMAD4* that forms a heterotrimer with *SMAD2* and *SMAD3* to exert transcriptional activity plays a crucial role in carcinogenesis [35], and loss of the *SMAD4* gene occurs in about 30% of CRC [36]. It was reported that loss of *SMAD4* was significantly related to CRC progression and metastasis and occurred in late stages [29,37,38,39,40]. Previous studies revealed that in colon cancer, the activation of TGF-β signaling induced ERK and P38 signaling and stimulated angiogenesis by *VEGF* upregulation when *SMAD4* was knocked down [36]. Our findings are in agreement with previous investigations [41,42] and indicate that the expression level of the *SMAD4* gene in stage III-IV CRC was lower than that in stage I-II, although this difference did not reach a statistically significant level. 

We observed an upregulation of the *TGF-β* gene in stage III-IV CRC samples, although these results were not statistically significant. According to these data, loss of the *SMAD4* gene and upregulation of *TGF-β* may be associated with a more advanced tumor stage and can promote cancer initiation, progression, and metastasis [35]. 

The NOTCH signaling pathway activity has been reported in several cancers, such as CRC and hepatocarcinoma [43,44,45,46]. NOTCH signaling consists of several receptors (NOTCH1–4) and targets genes including the transcription factor *HES1* (HES family basic helix-loop-helix (bHLH) transcription factor 1) [43,47,48]. In addition, *HES1* has several functions, such as intestinal cell stability and apoptosis control [43,49]. Our study showed a downregulation of *NOTCH1* in stage III-IV CRC and a significant correlation between a high expression level of *NOTCH1* and longer patient survival. This is in contrast with a previous study that reported an upregulation of *NOTCH1* in advanced or metastatic CRC patients and a significant association between the upregulation of *NOTCH1* and poor survival [43]. Moreover, we observed a significant overexpression of *NOTCH1* levels in female patients.

In agreement with our findings, several studies demonstrated a significant overexpression of the *HES1* gene in CRC samples compared with a normal tissue [50,51,52]. Additionally, we found that the overexpression of the *HES1* gene in stage III-IV CRC was more marked than in stage I-II. This result is in agreement with past investigations [43,50,52]. Overall, our data, together with those of others, confirm that NOTCH signaling, especially *NOTCH1* and *HES1*, play a critical role in metastasis and invasion as well as in the activation of several other signaling pathways. The activation of *NOTCH1* is, in fact, able to induce *HES1* and to start cancer progression [53,54,55,56]. 

*IL2RA* and *IL2RB* bind interleukin 2, which is necessary for the stimulation of T-cell immune response, and act as signal transduction factors. A high level of *IL2*, *IL2RA*, and *IL2RB* gene expression and their relationships with tumor progression and malignancy have been previously reported [57]. However, in agreement with our results, Marshall et al. also found no significant association between *IL2RB* gene expression and cancer progression [58].In the present investigation, we observed that a downregulation of *IL2RA* was significantly associated with CRC patients younger than 50 years. 

*ANXA3* plays a relevant role in tumor metastasis, invasion, and drug treatment resistance [59,60,61,62]. In a previous investigation, a blood-based biomarker panel, also including the *ANXA3* gene, able to stratify subjects according to their relative CRC risk in comparison with an average-risk population, was developed [58]. Similar to this study, a significant upregulation of *ANXA3* has been identified in CRC tissues compared with normal mucosa as well as in several other cancers, such as pancreas, breast, and lung cancers [63,64]. 

Several findings have shown that the suppression of *ANXA3* upregulation could inhibit cell proliferation and metastasis in CRC [65,66]. Thus, *ANXA3* could be considered a new potential prognostic biomarker and therapeutic target for CRC treatment [66,67]. Upregulation of the *ANXA3* gene and its correlation with gastric tumor size, stage, and LNMs were detected by Wang et al. [67] Moreover, these authors suggested that the overexpression of ANXA3 has a huge effect on gastric cancer malignancy, and it can be used as a novel prognostic biomarker and a suitable target for treatment [67]. Zhou et al. reported that high expression levels of ANXA3 were significantly correlated with breast tumor LNMs and tumor grade, suggesting ANXA3 as a biomarker for breast cancer prognosis [68]. In our study, a significant overexpression of the *ANXA3* gene in well-differentiated tumors compared with poorly differentiated tumors was instead observed. However, we also observed higher levels of *ANXA3* in tumors from patients with LNMs compared with tumors from patients without LNMs, although this difference did not reach statistical significance. Overall, we were only able to partially confirm the observations of other authors who suggested that the *ANXA3* gene may act as an oncogene and play a role in the transformation of a normal tissue into tumor, CRC invasion, and malignancy progression.

*BUB1* acts as a checkpoint factor during cell mitosis and proliferation, and *PCSK7* plays a role in cellular multiplication, mortality, and adhesion. These two genes are involved in cancer metastasis and invasion [69,70,71,72]. In our study, we observed a downregulation of the *BUB1* gene in CRC samples compared with normal mucosa, but this difference was not statistically significant. Additionally, previous studies showed a downregulation of the *BUB1* gene in gastric cancer and in CRC [70,73]. Furthermore, in agreement with previous studies, we found an upregulation of *PCSK7* in CRC. On the other hand, Jaaks et al. realized a significant upregulation of the *PCSK7* gene in colon cancer and considered it a potential biomarker [72]. Taken together, it could be suggested that the low expression level of *BUB1* and the upregulation of *PCSK7* have a critical role in malignant transition through colorectal carcinogenesis.

EGFR is a transmembrane receptor that binds to EGF and stimulates cell growth in tissues. Overexpression of the *EGFR* gene has been observed in several cancers, including colorectal, lung, breast, and bladder [74]. It has been reported that the *EGFR* gene may play a role in CRC development [75] since its expression increases with malignant transformation from normal colonic mucosa to metastatic CRC [75,76]. Previous studies, in fact, showed that *EGFR* gene overexpression was significantly related to tumor stages, metastasis, and well-differentiated tumors [75,77]. Although we found a higher expression of *EGFR* in tumor tissues compared with normal tissues, as well as in the right colon compared with the left colon, we did not find correlations between tumor *EGFR* gene expression and clinicopathological features. However, our observations substantially confirm the role of EGFR in CRC progression. 

## 5. Conclusions

In conclusion, the main results of this study highlight that the expression of the tumor *VANGL1* gene is an independent prognostic biomarker and could be considered a potential predictor for detecting malignancy risk in CRC patients. Additionally, LNMs were highly predicted by the 20-gene panels. However, validation studies including a higher number of patients are required.

## Figures and Tables

**Figure 1 jpm-11-00126-f001:**
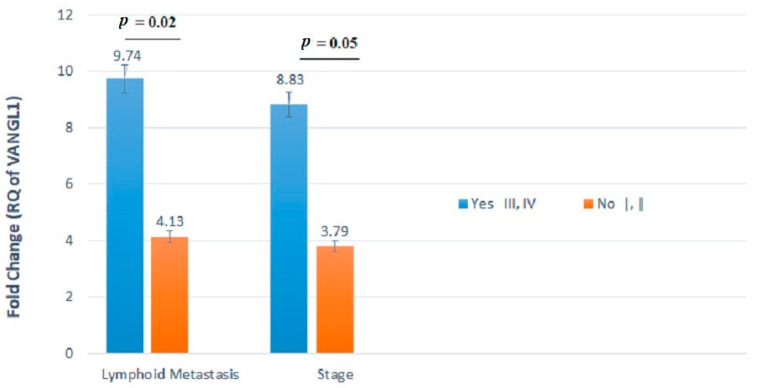
*VANGL1* mRNA relative quantification (RQ) established by RT-qPCR analysis according to lymph node metastasis (LNMs) involvement or stage. Gene expression levels of *VANGL1* differed significantly between patients with and without LNMs.

**Figure 2 jpm-11-00126-f002:**
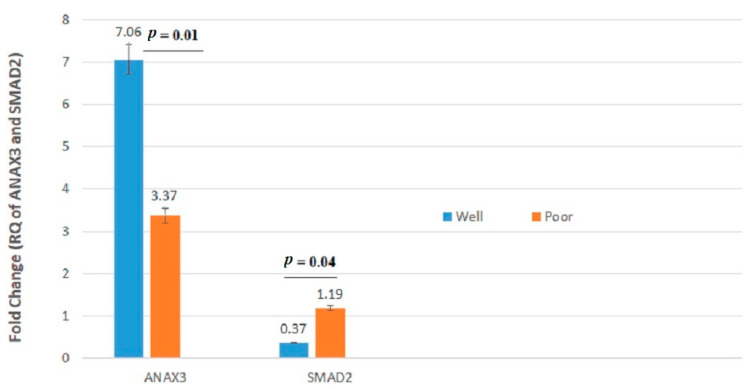
*ANXA3* and *SMAD2* mRNA relative quantification (RQ) established by RT-qPCR analysis according to histological grade. Gene expression levels of these genes differed significantly between well- and poorly differentiated cancers.

**Figure 3 jpm-11-00126-f003:**
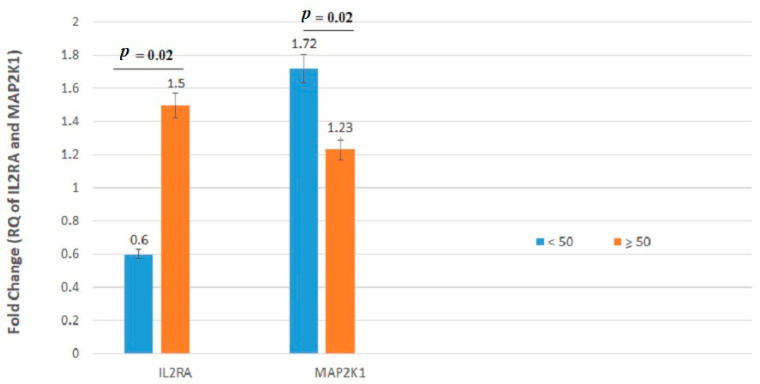
*IL2RA* and *MAP2K1* mRNA relative quantification established by RT-qPCR analysis according to age. Gene expression levels of these genes differed significantly between patients younger than 50 years and patients older than or equal to 50 years.

**Figure 4 jpm-11-00126-f004:**
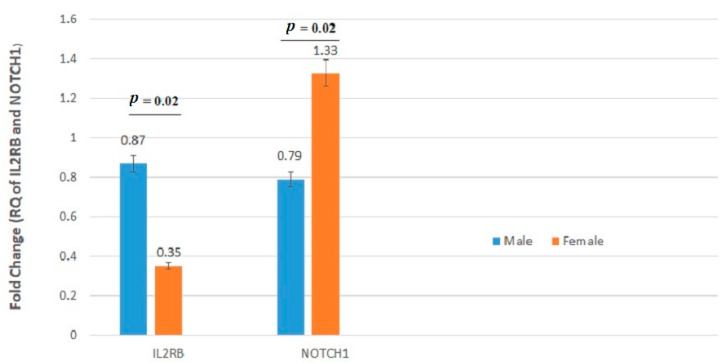
*IL2RB* and *NOTCH1* mRNA relative quantification (RQ) established by RT-qPCR analysis according to gender. Gene expression levels of these genes differed significantly between males and females.

**Figure 5 jpm-11-00126-f005:**
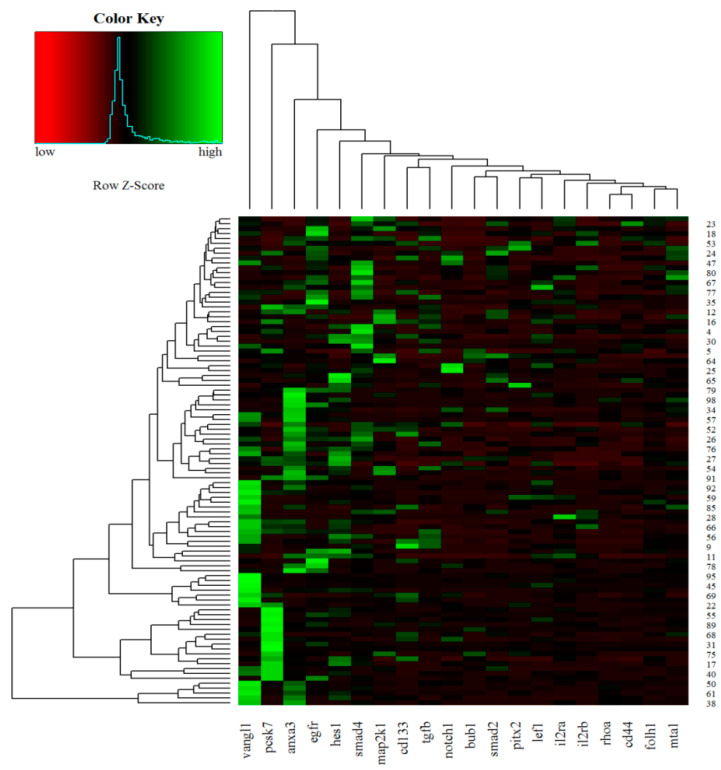
Heat maps of real-time RT-qPCR data representing gene expression variations of the 20 analyzed transcripts in FFPE CRC samples. Green indicates upregulation, and red indicates downregulation.

**Figure 6 jpm-11-00126-f006:**
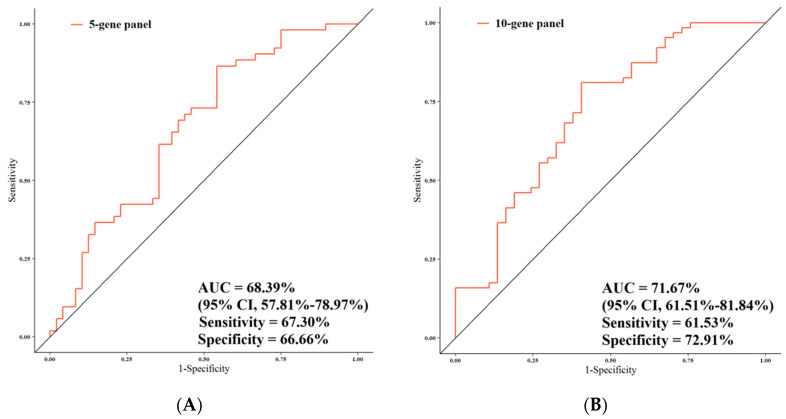
Comparison of the predictive performance by receiver operating characteristic (ROC) curve analysis for stage. (**A**) The AUC assessment of the logit(*p*) value for the panel of 5 genes. (**B**) The AUC assessment of the logit(*p*) value for the panel of 10 genes. (**C**) The AUC assessment of the logit(*p*) value for the panel of total genes. (**D**) Comparison of the predictive performance for the panel of 5, 10, and 20 genes.

**Figure 7 jpm-11-00126-f007:**
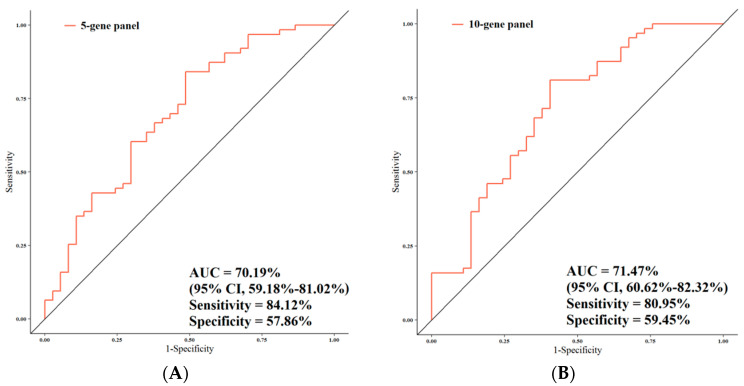
Comparison of the predictive performance by receiver operating characteristic (ROC) curve analysis for lymph node metastasis (LNMs). (**A**) The AUC assessment of the logit(*p*) value for the panel of 5 genes. (**B**) The AUC assessment of the logit(*p*) value for the panel of 10 genes. (**C**) The AUC assessment of the logit(*p*) value for the panel of total genes. (**D**) Comparison of the predictive performance for the panel of 5, 10, and 20 genes.

**Figure 8 jpm-11-00126-f008:**
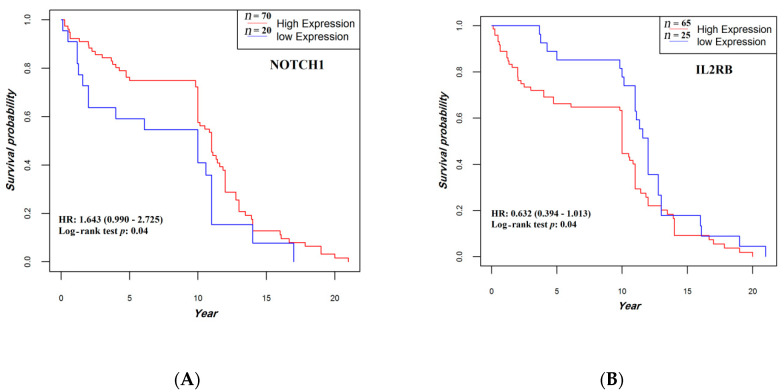
Association between *NOTCH1* (**A**) and *IL2RB* (**B**) expression and overall survival. The median follow-up was 10.8 (0.95LCL–0.95UCL; 10–11) years. HR, hazard ratio.

**Figure 9 jpm-11-00126-f009:**
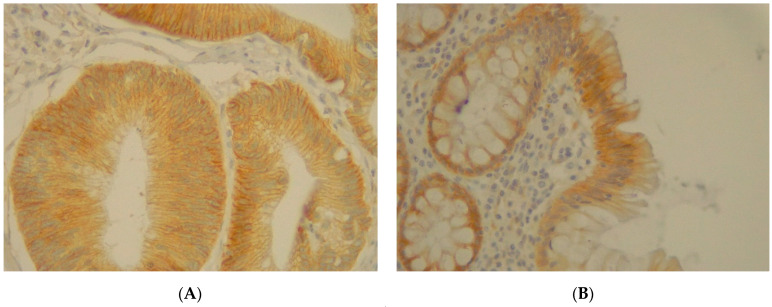
(**A**) Immunohistochemical staining of VANGL1 in colorectal adenocarcinoma with an intensity score of 3+ with more than 70% of the stained cells. (**B**) Immunohistochemical staining of VANGL1 in normal sample.

**Figure 10 jpm-11-00126-f010:**
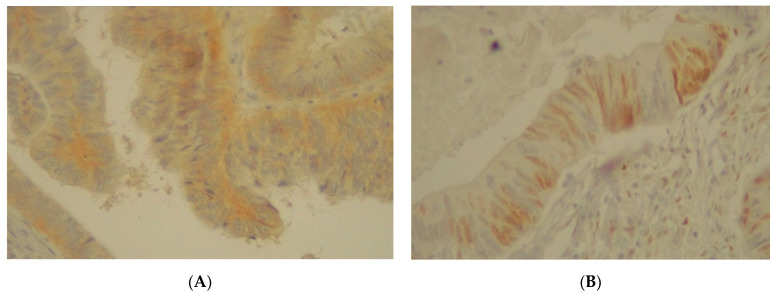
(**A**) Immunohistochemical staining of EGFR in colorectal adenocarcinoma with an intensity score of 2+ with more than 50% of the stained cells. (**B**) Immunohistochemical staining of LEF1 in colorectal adenocarcinoma with a positive expression of more than 20% of the stained cells.

**Figure 11 jpm-11-00126-f011:**
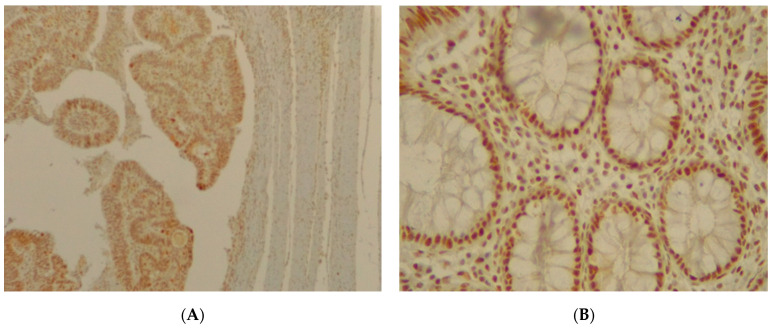
(**A**) Immunohistochemical staining of SMAD4 in colorectal adenocarcinoma with an intensity score of 2+ with more than 50% of the stained cells. (**B**) Immunohistochemical staining of SMAD4 in normal sample.

**Table 1 jpm-11-00126-t001:** Information * on the biological functions of 20 candidate genes and on the primer sequences used for RT-qPCR

Gene Symbol	Gene Name and Functions	Primer Sequence
*VANGL1* *(KITENIN)*	VANGL planar cell polarity protein 1 (located on chromosome 1) 1. It encodes a member of the tetraspanin family. 2. It may be involved in mediating intestinal trefoil factor-induced wound healing in the intestinal mucosa.	F: 5′-GACACAAGTCACCCCGGAATA-3′ R: 5′-TCCTCTGTCCGAGTAGAATCATT-3′ Amplicon length: 109 bp
*IL2RA* *(CD25)*	Interleukin 2 receptor subunit alpha (located on chromosome 10) 1. Mutations in this gene are associated with interleukin 2 receptor alpha deficiency. 2. Serum IL-2R levels are found to be elevated in patients with different types of carcinomas.	F: 5′-GAACACAACGAAACAAGTGACAC-3′ R: 5′-GGCTGCATTGGACTTTGCATT-3′ Amplicon length: 81 bp
*IL2RB* *(CD122)*	Interleukin 2 receptor subunit beta (located on chromosome 22) 1. It is involved in receptor-mediated endocytosis and transduction of mitogenic signals from IL2.	F: 5′-CAGCGGTGAATGGCACTTC-3′ R: 5′-GGCATGGACTTGGCAGGAA-3′ Amplicon length: 113 bp
*TGFβ1*	Transforming growth factor ß (located on chromosome 19) 1. It encodes a secreted ligand of the TGFß superfamily of proteins. The encoded protein regulates cell proliferation, differentiation, and growth. 2. It is frequently upregulated in tumor cells.	F: 5′-CAATTCCTGGCGATACCTCAG-3′ R: 5′-GCACAACTCCGGTGACATCAA-3′ Amplicon length: 86 bp
*SMAD2*	SMAD2 family member 2 (located on chromosome 18) 1. The protein encoded mediates the signal of the transforming growth factor (TGF)-beta, thus regulating multiple cellular processes (i.e., cell proliferation, apoptosis, and differentiation).	F: 5′-CCGACACACCGAGATCCTAAC-3′ R: 5′-GAGGTGGCGTTTCTGGAATATAA-3′ Amplicon length: 125 bp
*SMAD4*	SMAD4 family member 4 (Located on chromosome 18) 1. The encoded protein forms homomeric complexes and heteromeric complexes with other activated Smad proteins that accumulate in the nucleus and regulate the transcription of target genes. 2. The encoded protein acts as a tumor suppressor and inhibits epithelial cell proliferation.	F: 5′-CCACCAAGTAATCGTGCATCG-3′ R: 5′-TGGTAGCATTAGACTCAGATGGG-3′ Amplicon length: 76 bp
*CD44* *(CSPG8)*	CD44 antigen (located on chromosome 11) 1. It is a cell-surface glycoprotein involved in cell–cell interactions, cell adhesion, and migration. 2. The encoded protein participates in several cellular functions (e.g., lymphocyte activation, recirculation and homing, hematopoiesis, and tumor metastasis).	F: 5′-CTGCCGCTTTGCAGGTGTA-3′ R: 5′-CATTGTGGGCAAGGTGCTATT-3′ Amplicon length: 109 bp
*CD133* *(PROM1)*	CD133 (located on chromosome 4) 1. It encodes a pentaspan transmembrane glycoprotein. 2. The encoded protein is often expressed on adult stem cells, where it has been suggested to maintain stem cell properties by suppressing differentiation. 3. It is the marker most commonly used for the isolation of cancer stem cell population from different tumors.	F: 5′-GGCCCAGTACAACACTACCAA-3′ R: 5′-ATTCCGCCTCCTAGCACTGAA-3′ Amplicon length: 75 bp
*HES1*	HES family basic helix-loop-helix (bHLH) transcription factor 1 (chromosome 3) 1. It is a transcriptional repressor of genes that require a bHLH protein for their transcription. 2. It plays an important role in the Notch signaling pathway. 3. The absence of Hes1 in the developing intestine promotes the increase of Math1 (the production of intestinal cell types).	F: 5′-ACGTGCGAGGGCGTTAATAC-3′ R: 5′-GGGGTAGGTCATGGCATTGA-3′ Amplicon length: 90 bp
*NOTCH1*	NOTCH receptor 1 (located on chromosome 9) 1. The Notch signaling pathway regulates interactions between physically adjacent cells through the binding of Notch family receptors to their cognate ligands. 2. It plays a role in the development of numerous cell and tissue types. 3. Mutations in *NOTCH1* are associated with syndromes, hematological and solid tumors.	F: 5′-TGGACCAGATTGGGGAGTTC-3′ R: 5′-GCACACTCGTCTGTGTTGAC-3′ Amplicon length: 82 bp
*LEF1* *(TCF10)*	Lymphoid enhancer-binding factor 1 (located on chromosome 4) 1. It encodes a transcription factor belonging to a family of proteins that share homology with the high mobility group protein-1. 2. It binds to a functionally important site in the T-cell receptor-alpha (TCRA) enhancer, thus conferring maximal enhancer activity. 3. It is involved in the Wnt signaling pathway, and mutations in this gene have been found in some tumors.	F: 5′-ATGTCAACTCCAAACAAGGCA-3′ R: 5′-CCCGGAGACAAGGGATAAAAAGT-3′ Amplicon length: 76 bp
*MTA1*	Metastasis-associated 1 (located on chromosome 14) 1. MTA1 expression has been correlated with the metastatic potential of some carcinomas, but it is expressed also in many normal tissues. 2. The profile and activity of the encoded protein suggest that it is involved in regulating transcription and that this may be accomplished by chromatin remodeling.	F: 5′-ACGCAACCCTGTCAGTCTG-3′ R: 5′-GGGCAGGTCCACCATTTCC-3′ Amplicon length: 104 bp
*EGFR* *(ErBb-1)*	Epidermal growth factor receptor (located on chromosome 7) 1. It encodes a transmembrane glycoprotein that is a member of the protein kinase (PK) superfamily. 2. EGFR binds to EGF, thus inducing receptor dimerization and tyrosine autophosphorylation, leading to cell proliferation. 3. Mutations in EGFR are associated with lung cancer.	F: 5′-TGCGTCTCTTGCCGGAAT-3′ R: 5′-GGCTCACCCTCCAGAAGGTT-3′ Amplicon length: 71 bp
*MAP2K1* *(MEK1)*	Mitogen-activated protein kinase kinase 1 (located on chromosome 15) 1. The encoded protein is a member of the dual specificity PK family that acts as a mitogen-activated protein (MAP) kinase kinase. 2. The encoded protein stimulates the enzymatic activity of MAP kinases upon a wide variety of extra- and intracellular signals. 3. As a component of the MAP kinase signal transduction pathway, the encoded protein is involved in many cellular processes (e.g., proliferation, differentiation, and transcription regulation).	F: 5′-CAATGGCGGTGTGGTGTTC-3′ R: 5′-GATTGCGGGTTTGATCTCCAG-3′ Amplicon length: 91 bp
*FOLH1* *(PSMA)*	Folate hydrolase 1 (located on chromosome 11) 1. It encodes a type II transmembrane glycoprotein belonging to the M28 peptidase family. 2. Also known as prostate-specific membrane antigen (PSMA), it is expressed in many tissues, including the prostate. 3. In the prostate, the FOLH1/PSMA protein is upregulated in cancer cells and is used as an effective diagnostic and prognostic indicator of prostate cancer.	F: 5′-AGAGGGCGATCTAGTGTATGTT-3′ R: 5′-TGATTTTCATGTCCCGTTCCAAT-3′ Amplicon length: 74 bp
*BUB1*	BUB1 mitotic checkpoint serine/threonine kinase (located on chromosome 2) 1. It encodes a protein that plays a central role in mitosis. 2. This protein may also function in the DNA damage response. 3. Mutations in this gene have been associated with aneuploidy and several forms of cancer.	F: 5′-AGCCCAGACAGTAACAGACTC-3′ R: 5′-GTTGGCAACCTTATGTGTTTCAC-3′ Amplicon length: 136 bp
*RHOA*	Ras homolog family member A (located on chromosome 3) 1. It encodes a member of the Rho family of small GTPases that function as molecular switches in signal transduction cascades. 2. Overexpression of this gene is associated with tumor cell proliferation and metastasis.	F: 5′-GGAAAGCAGGTAGAGTTGGCT-3′ R: 5′-GGCTGTCGATGGAAAAACACAT-3′ Amplicon length: 118 bp
*PCSK7*	Proprotein convertase subtilisin/kexin type 7 (located on chromosome 11) 1. It encodes a type 1 membrane-bound protease that is expressed in many tissues, including the neuroendocrine, liver, gut, and brain. 2. It has been implicated in the transcriptional regulation of housekeeping genes. 3. A chromosomal translocation associated with B-cell lymphoma occurs between this gene and its inverted counterpart.	F: 5′-GCAGCGTCCACTTCAACGA-3′ R: 5′-GCCCAGTCACATTGCGTTC-3′ Amplicon length: 117 bp
*PITX2* *(ARP1)*	Paired-like homeodomain 2 (located on chromosome 4) 1. It encodes a member of the RIEG/PITX homeobox family, which is in the bicoid class of homeodomain proteins. 3. The encoded protein acts as a transcription factor and regulates procollagen lysyl hydroxylase gene expression.	F: 5′-GCCAAGGGCCTTACATCCG-3′ R: 5′-GGTGGGGAAAACATGCTCTG-3′ Amplicon length: 101 bp
*ANXA3* *(annexin) A3*	Annexin A3 (located on chromosome 4) 1. It encodes a member of the annexin family. 1. Members of this calcium-dependent phospholipid-binding protein family play a role in the regulation of cellular growth and in signal transduction pathways. 2. The encoded protein functions in the inhibition of phospholipase A2 and cleavage of inositol 1,2-cyclic phosphate to form inositol 1-phosphate.	F: 5′-TTAGCCCATCAGTGGATGCTG-3′ R: 5′-CTGTGCATTTGACCTCTCAGT-3′ Amplicon length: 104 bp
*B2M*	β2-microglobulin (located on chromosome 15) – Housekeeping gene	F: 5′-TGCTGTCTCCATGTTTGATGTATCT-3′ R: 5′-TCTCTGCTCCCCACCTCTAAGT-3′ Amplicon length: 86 bp
*ACTB*	β-actin (located on chromosome 7) – Housekeeping gene	F: 5′- GCCGGGACCTGACTGACTAC-3′ R: 5′- TTCTCCTTAATGTCACGCACGAT-3′ Amplicon length: 100 bp

* Information available at https://www.ncbi.nlm.nih.gov/ (accessed on 10 February 2021).

**Table 2 jpm-11-00126-t002:** Clinical and pathological characteristics of patients.

Characteristics	Number of Patients, 100 (%)
Gender	
Male	59 (59%)
Female	41 (41%)
Age	
<50 years	59 (59%)
≥50 years	41 (41%)
Tumor localization	
Left (descending colon)	49 (49%)
Right (ascending colon)	51 (51%)
Tumor stage	
I	9 (9%)
II	43 (43%)
ΙIΙ	41 (41%)
IV	7 (7%)
Differentiation grade	
Well differentiated	82 (82%)
Poorly differentiated	18 (18%)
Lymph node metastasis	
Yes	37 (37%)
No	63 (63%)
Median overall survival, range	10.8 years, 0.019–21 years

**Table 3 jpm-11-00126-t003:** Relationships between the expression of 20 CRC study genes and age and sex.

Gene Symbol	Fold Change of Age			Fold Change of Sex		
	<50	≥50	*p*-Value	Male	Female	*p*-Value
*VANGL1*	4.910 ± 7.56	7.155 ± 10.54	0.44	5.706 ± 9.02	6.972 ± 10.08	0.36
*IL2RA*	0.603 ± 0.640	1.509 ± 1.83	0.02	1.148 ± 1.61	1.098 ± 1.37	0.93
*IL2RB*	0.699 ± 1.10	0.635 ± 1.24	0.88	0.870 ± 1.37	0.351 ± 0.71	0.02
*TGFβ*	1.056 ± 1.61	1.285 ± 2.02	0.79	1.308 ± 2.14	1.010 ± 1.34	0.90
*SMAD2*	1.313 ± 2.20	0.855 ± 1.45	0.74	0.942 ± 1.64	1.205 ± 2.06	0.30
*SMAD4*	2.157 ± 2.36	2.425 ± 2.21	0.41	2.417 ± 2.20	2.157 ± 2.37	0.34
*CD44*	0.596 ± 0.814	0.537 ± 0.87	0.44	0.478 ± 0.85	0.687 ± 0.82	0.10
*CD133*	0.832 ± 1.77	1.329 ± 2.83	0.62	0.870 ± 2.43	1.495 ± 2.44	0.07
*HES1*	2.830 ± 3.52	2.134 ± 3.28	0.36	2.685 ± 3.39	0.955 ± 3.38	0.44
*NOTCH1*	1.088 ± 2.18	0.953 ± 2.22	0.58	0.795 ± 2.36	1.332 ± 1.89	0.02
*LEF1*	1.178 ± 1.80	1.038 ± 1.19	0.58	0.988 ± 1.14	1.260 ± 1.86	0.86
*MTA1*	1.045 ± 1.01	1.012 ± 0.98	0.85	0.947 ± 0.93	1.144 ± 1.06	0.37
*EGFR*	2.272 ± 2.60	2.937 ± 4.65	0.75	2.439 ± 3.43	2.986 ± 4.58	0.87
*MAP2K1*	1.722 ± 2.47	1.235 ± 2.23	0.02	1.495 ± 2.65	1.355 ± 1.78	0.49
*FOLH1*	0.797 ± 0.70	0.814 ± 0.91	0.65	0.809 ± 0.85	0.804 ± 0.81	0.78
*BUB1*	0.656 ± 1.65	0.610 ± 1.46	0.18	0.731 ± 1.53	0.476 ± 1.56	0.27
*RHOA*	0.510 ± 0.621	0.562 ± 1.19	0.11	0.625 ± 1.11	0.414 ± 0.75	0.50
*PCSK7*	6.590 ± 7.74	3.069 ± 4.48	0.19	4.196 ± 5.12	5.076 ± 6.02	0.42
*PITX2*	0.976 ± 2.00	0.663 ± 1.13	0.57	0.720 ± 1.13	0.906 ± 2.05	0.58
*ANXA3*	3.533 ± 4.70	4.399 ± 6.65	0.60	4.298 ± 6.06	3.641 ± 5.71	0.12

**Table 4 jpm-11-00126-t004:** Relationships between the expression of 20 CRC study genes and clinical and pathological characteristics of patients.

Gene Symbol	Fold Change of Stage			Fold Change of Lymph Node Metastasis			Fold Change of Tumor Site			Fold Change of Differentiation Grade		
	I, II	ΙIΙ, IV	*p*-Value	Yes	No	*p*-Value	Right	Left	*p*-Value	Well	Poor	*p*-Value
*VANGL1*	3.795 ± 5.63	8.831 ± 11.80	0.05	9.749 ± 12.40	4.136 ± 6.38	0.02	7.144 ± 10.44	5.243 ± 8.23	0.26	6.516 ± 9.22	4.831 ± 10.50	0.07
*IL2RA*	1.073 ± 1.59	1.188 ± 1.44	0.40	0.982 ± 1.19	1.214 ± 1.68	0.85	1.293 ± 1.84	0.957 ± 1.08	0.63	1.221 ± 1.56	0.707 ± 1.23	0.12
*IL2RB*	0.728 ± 1.31	0.591 ± 1.02	0.64	0.433 ± 0.43	0.797 ± 1.33	0.19	0.520 ± 0.85	0.811 ± 1.44	0.78	0.618 ± 1.21	0.864 ± 1.00	0.05
*TGFβ*	0.977 ± 1.50	1.418 ± 2.17	0.71	1.560 ± 2.25	0.971 ± 1.56	0.22	1.086 ± 1.66	1.296 ± 2.06	0.53	1.649 ± 1.84	1.088 ± 1.90	0.26
*SMAD2*	1.257 ± 2.13	0.821 ± 1.37	0.27	0.702 ± 1.26	1.250 ± 2.05	0.19	1.053 ± 2.17	1.043 ± 1.36	0.24	0.371 ± 1.95	1.196 ± 0.65	0.04
*SMAD4*	2.483 ± 2.50	2.128 ± 1.99	0.78	2.096 ± 1.99	2.440 ± 2.41	0.82	2.165 ± 2.41	2.467 ± 2.11	0.37	2.068 ± 2.20	2.367 ± 2.60	0.22
*CD44*	0.480 ± 0.61	0.650 ± 1.03	0.77	0.767 ± 1.14	0.441 ± 0.58	0.65	0.705 ± 1.03	0.413 ± 0.55	0.24	0.570 ± 0.76	0.522 ± 1.15	0.35
*CD133*	0.842 ± 1.99	1.421 ± 2.85	0.63	1.324 ± 2.43	1.000 ± 2.46	0.50	0.831 ± 1.59	1.421 ± 3.08	0.45	1.204 ± 2.63	0.738 ± 1.21	0.26
*HES1*	2.111 ± 2.96	2.767 ± 3.80	0.44	2.944 ± 4.06	2.123 ± 2.91	0.33	2.707 ± 3.90	2.134 ± 2.76	0.29	2.637 ± 3.58	1.465 ± 2.11	0.47
*NOTCH1*	1.147 ± 2.44	0.861 ± 1.90	0.66	0.968 ± 2.10	1.034 ± 2.26	0.87	0.768 ± 1.49	1.261 ± 2.73	0.16	1.170 ± 2.38	0.278 ± 0.49	0.13
*LEF1*	0.844 ± 0.87	1.371 ± 1.89	0.76	1.424 ± 1.99	0.905 ± 1.02	0.67	0.980 ± 1.49	1.218 ± 1.45	0.21	1.056 ± 1.34	1.283 ± 2.00	0.51
*MTA1*	0.983 ± 0.98	1.072 ± 1.00	0.72	1.073 ± 1.03	0.998 ± 0.96	0.83	1.178 ± 0.97	0.868 ± 0.99	0.08	1.024 ± 0.98	1.032 ± 1.04	0.89
*EGFR*	2.693 ± 3.86	2.620 ± 4.02	0.84	2.091 ± 2.65	2.990 ± 4.49	0.56	3.214 ± 4.31	2.078 ± 3.40	0.22	2.595 ± 4.12	2.945 ± 2.91	0.33
*MAP2K1*	1.442 ± 2.57	1.436 ± 2.07	0.33	1.549 ± 1.25	1.253 ± 2.58	0.98	1.251 ± 2.09	1.636 ± 2.57	0.34	1.377 ± 2.34	1.724 ± 2.34	0.39
*FOLH1*	0.857 ± 0.90	0.753 ± 0.75	0.55	0.767 ± 0.76	0.831 ± 0.87	0.79	0.848 ± 0.90	0.764 ± 0.75	0.70	0.829 ± 0.86	0.708 ± 0.67	0.83
*BUB1*	0.607 ± 1.35	0.652 ± 1.73	0.95	0.617 ± 1.67	0.636 ± 1.47	0.96	0.559 ± 1.28	0.702 ± 1.78	0.65	0.695 ± 1.68	0.328 ± 0.48	0.27
*RHOA*	0.294 ± 0.39	0.807 ± 1.32	0.21	0.916 ± 1.46	0.320 ± 0.43	0.24	0.645 ± 1.13	0.431 ± 0.81	0.36	0.501 ± 0.86	0.720 ± 1.44	0.78
*PCSK7*	4.290 ± 5.74	4.827 ± 6.81	0.12	4.170 ± 5.51	4.770 ± 6.21	0.19	3.614 ± 4.51	5.520 ± 6.32	0.07	4.364 ± 5.65	5.384 ± 6.96	1.00
*PITX2*	0.644 ± 1.03	0.957 ± 1.95	0.98	1.079 ± 2.16	0.627 ± 1.03	0.63	0.959 ± 1.98	0.622 ± 0.92	0.84	0.737 ± 1.12	1.056 ± 2.82	0.21
*ANXA3*	3.320 ± 4.54	4.811 ± 7.06	0.67	5.542 ± 7.75	3.150 ± 4.31	0.21	4.597 ± 7.06	3.450 ± 4.39	0.73	7.065 ± 4.91	3.370 ± 8.71	0.01

## Data Availability

The data presented in this study are available on request from the corresponding author.

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
