# Peer review of "Increased Expression of VANGL1 is Predictive of Lymph Node Metastasis in Colorectal Cancer: Results from a 20-Gene Expression Signature"

_jpm, 2021, doi:10.3390/jpm11020126_

Round 1

Reviewer 1 Report

Several grammatical errors were noted:

line 80: is this sentence referring to chemotherapy and radiation naiive samples?

line 89: extra punctuation

line 168: #s are not consistent

line 190: demographic rather than demographically

line 202: older rather than oldest

line 420: consider revision, not sure what this is saying

Some clarification in figures would be helpful. Specifically,

Table 2: is it necessary to included the combined numbers when you have included the number of cases for each stage?

Table 3: very difficult to read. Possibly consider presenting table in landscape format to improve clarity

Figure 1: consider revision of figure legend to clarify

For the IHC studies, suggest consideration of further explanation to justify the selection of genes based on the expression analysis. All of these genes are clearly important in CRC treatment and pathogenesis but an explanation of how they relate to your quantitative expression analysis would be beneficial to their relevance to the other work presented in the paper. 

Author Response

Reply letter

R 1:

We wish to thank very much the reviewer for the valuable suggestions.

Q1: line 80: is this sentence referring to chemotherapy and radiation naiive samples?

R1: Yes, all patients were chemo- and radiotherapy naïve. We have now specified this in the sentence

Q2: line 89: extra punctuation

R2: Thank you. We have deleted the full stop between “were” and “patients” and we have modified the sentence as follows: “The Exclusion criteria were the following:”

Q3: line 168: #s are not consistent

Q4: Thank you. We have deleted the ambiguous sentence

Q5: line 190: demographic rather than demographically

R5: Thank you. We have replaced “demographically” with “demographic”

Q6: line 202: older rather than oldest

R6: Thank you. We have replaced “oldest” with “older”

Q7: line 420: consider revision, not sure what this is saying

Q8: Thank you. We agree with the referee that the sentence was not consistent with the obtained data. Thus, we have modified it in agreement with our results and by recontextualizing our data within those available from the literature

Q9 : Some clarification in figures would be helpful. Specifically,

R9: Thank you. The legends of the figures have been improved

Q10: Table 2: is it necessary to included the combined numbers when you have included the number of cases for each stage?

R10: Thank you. We followed your suggestion and combined numbers for stage have been deleted

Q11: Table 3: very difficult to read. Possibly consider presenting table in landscape format to improve clarity

R11: Thank you. Due to the high number of genes as well as the wide differences among numbers reported (e.g. from 0.1 to more than 10) and also the need to include p value for each gene and comparison in order to be scientifically correct, in our opinion, a graphical representation of this table would reduce its informative content. Thus, we have thought to obtain two tables. The first one with comparisons concerning age and sex, the second one with comparisons between clinical/pathological variables and gene expression. We understand that this solution does not exactly satisfy the request of the reviewer but we hope that it can be accepted.

Q12: Figure 1: consider revision of figure legend to clarify

R12: Thank you. The legends of the figures have been improved

Q13: For the IHC studies, suggest consideration of further explanation to justify the selection of genes based on the expression analysis. All of these genes are clearly important in CRC treatment and pathogenesis but an explanation of how they relate to your quantitative expression analysis would be beneficial to their relevance to the other work presented in the paper. 

R13: Thank you. We have now added further information on the reason for which the 4 study proteins were selected

Reviewer 2 Report

It is a very interesting genetic study with promising results regarding possible prediction of lymph node metasthasis in colorectal cancer. 

Please define FFPE as it first appears in the abstract (raw 31). 

Raws 86 and 89: precised the meaning of term "main". Where there other criteria? Which? Otherwise exclude the word "main". 

Raw 89: replace "." with ":" - "Exclusion criteria were:"

There is a dicordance regarding splitting the patients according to their age between the affirmation in raw 156 (<55; >=55) and the rest of the study (results, tables 2, 3). You have to correct that. 

Raws 236-237: the word "genes" is repeted twice. 

Raws 371, 403, 423, 431: allign the indentation.

Author Response

R2:

We wish to thank very much the reviewer for the valuable suggestions.

Q1: Please define FFPE as it first appears in the abstract (raw 31). 

R1: Thank you. FFPE has been now abbreviated the first time it appears

Q2: Raws 86 and 89: precised the meaning of term "main". Where there other criteria? Which? Otherwise exclude the word "main". 

R2: Thank you. We deleted the term “main” from inclusion and exclusion criteria since those reported are the criteria considered

Q3: Raw 89: replace "." with ":" - "Exclusion criteria were:"

R3: Thank you. We have deleted the full stop between “were” and “patients” and we have modified the sentence as follows: “The Exclusion criteria were the following:”

Q4: There is a dicordance regarding splitting the patients according to their age between the affirmation in raw 156 (<55; >=55) and the rest of the study (results, tables 2, 3). You have to correct that. 

R4: Thank you. We have corrected the text since data reported in Tables are corrected.

Q5: Raws 236-237: the word "genes" is repeted twice. 

R5: Thank you. We have deleted one of the two “genes” term.

Q6: Raws 371, 403, 423, 431: allign the indentation.

R6: Thank you. The indentation of lines has been aligned

Round 2

Reviewer 1 Report

Thank you for your revisions.